# DYNAMIC SEMANTIC EQUIVALENCE CHECKING FOR ADVERSARIALLY ROBUST NEURAL COMPILERS

## ABSTRACT

We propose a novel architecture of neural compiler which incorporates a Dynamic Semantic Equivalence Checker (DSEC) to overcome the music of adversarial robustness in program compilation. Traditional neural compiler In both cases, traditional neural compilers are susceptible to the adversarial perturbations of input program code, leading to semantically incorrect optimization; the DSEC addresses this issue by using runtime verification when combined with probabilistic program analysis. The core innovation is the Relational Execution Tracker, a dynamic technique to compare execution traces of the original and compiled programs with the aid of a probabilistic divergence metric to identify behavioral differences. Furthermore, a Bayesian neural network-based Probabilistic Program Analyzer is used to assign perturbation likelihood estimate which allows targeted trace comparisons and efficient resource allocation. The system adaptively changes the optimization strategies in case of detection of adversarial influence, relying on a formally verified code generator for critical code regions. We also propose a hybrid adversarial training objective which constrains semantic consistency in addition to standard compilation accuracy.

## 1 INTRODUCTION

Neural compilers have emerged as powerful tools for optimizing program execution, transforming high-level code into efficient machine instructions through learned patterns rather than hand-crafted heuristics (Brauckmann et al., 2020). While these systems have shown impressive performance on common benchmarks, it has been poorly understood what the behavior of these systems is under adversarial conditions. Recent studies reveal that neural components in compilers can be vulnerable to subtle input perturbations that lead to incorrect optimizations or even security vulnerabilities (Yefet et al., 2020). This fragility creates great difficulties in deploying neural compilers for applications that demand safety and in which reliability is paramount.

The key challenge at the heart of the optimization versus robustness tension is therefore a special feature of neural compilation: there are inherent trade-offs in the optimization of neural compilation. Traditional compilers employ conservative transformations with formal correctness guarantees, whereas neural approaches often prioritize performance gains through data-driven optimizations (Li et al., 2020). Given malignly designed inputs, these neural optimizers could produce code that has a different meaning from that of the original program. Such deviations can propagate through compilation pipelines, potentially introducing subtle bugs or security flaws that evade conventional testing methodologies (Bielik & Vechev, 2020).

Existing approaches to adversarial robustness in neural networks primarily focus on classification tasks, leaving compiler-specific challenges largely unaddressed (Ganin et al., 2016). Techniques such as adversarial training and input preprocessing, while successful in vision or language models, do not take into account the structural and semantic constraint-based aspects of program compilation. Moreover, the dynamic nature of compiler optimizations requires robustness mechanisms that operate throughout the transformation pipeline rather than just at input boundaries (Leucker & Schallhart, 2009). This requires new ways to close the gap between neural program optimization and principles of formal verification.

We present a dynamic semantic equivalence checking framework that deals with the challenges following three key innovations. First, in our approach, we include runtime verification in the com-

pilation process itself, so we are continuously watching the behavioural correspondence from source to optimized programs. Second, we employ probabilistic program analysis to model uncertainty in both input perturbations and optimization outcomes, enabling robust decision-making under adversarial conditions (Chakarov & Sankaranarayanan, 2013). Third, the system implements relational reasoning principles to maintain semantic consistency across compilation stages, even when individual optimizations may appear locally valid but globally problematic (Santoro et al., 2017).

The proposed framework is also different from previous work in some very important respects. Unlike static verification methods that analyze programs in isolation, our dynamic approach captures runtime behaviors under actual execution conditions (Zhao et al., 2023). Compared to classical types of adversarial training, our method has additional privacy on the semantics of the programs that we want to preserve instead of just their output similarity. The integration of probabilistic analysis with relational reasoning provides theoretical grounding for robustness guarantees while remaining computationally tractable for practical compiler implementations (Eleftheriadis et al., 2022).

Our main contributions are: (1) a novel dynamic semantic equivalence checking mechanism for neural compilers combining runtime checking and probabilistic program checking; (2) a relational execution modeling tracking and comparing program behaviors at different compilation stages; (3) the adversarial training regime for compilation for semantic preservation particular in program optimization; and (4) extensive empirical validation proving improved robustness without compromising the overall compilation efficiency.

The rest of this paper is organized as it follows: Section 2 reviews related work in neural compilation and adversarial robustness. Section 3 gives needed background on the concepts of semantic equivalence checking and probabilistic program analysis. Section 4 describes our dynamic semantic equivalence checking framework. Experimental results are presented in Section 5 followed by discussion and future work in Section 6. We conclude in Section 7 with more general implications of such a research.

## 2 Related Work

The intersection between neural compilation and adversarial robustness covers several research lines, each of which discusses different parts of the challenge. We round up the work done on the problem in this area into three main categories: the type of semantic equivalence verification in compilers, adversarial robustness to program analysis models, and probabilistic methods to program verification.

### 2.1 Semantic Equivalence Verification in Compilers

Traditional compiler verification has for long relied on formal methods when ensuring semantic preservation through optimization passes. Recent work by Cheng et al. (2024) introduced denotational semantics for compiler verification, using Kleene algebra to establish equivalence between source and target programs. While mathematically rigorous, these approaches have difficulties scaling up and applying such learning to modern optimizing compilers with complex transformation sequences. The work on function-level equivalence checking (Malík & Vojnar, 2021) proposed algorithmic solutions for large-scale C projects, but their reliance on static analysis limits applicability to neural compilers where transformations are data-driven rather than rule-based.

Neural-specific equivalence checking has emerged more recently, with Dramko et al. (2025) developing techniques for neural decompilers. Their approach traces distributions of the variable values at time of execution, thus they guarantee equivalence with a some probability. However, this approach assumes benign inputs and does not consider adversarial perturbations. The DYCL framework (Chen et al., 2023) addressed semantic preservation in dynamic neural network compilation through program rewriting rules, but similarly lacks robustness considerations.

### 2.2 Adversarial Robustness for Program Analysis Models

The vulnerability of neural program analysis models to adversarial attacks has been demonstrated in several studies. Yefet et al. (2020) systematically analyzed attack surfaces in code representation models, showing that subtle syntactic changes can significantly alter model predictions. Subsequent

work by Bielik & Vechev (2020) explored adversarial training defenses, achieving modest robustness improvements through modified loss functions.

Specialized attacks against neural compilers were investigated in Liu et al. (2023b), which revealed that even correctly compiled programs could exhibit divergent runtime behaviors when subjected to adversarial inputs. Their results did encourage the focus of our work on dynamic rather than static verification. The CREATE framework (Ding et al., 2024) demonstrated that robustness improvements for one task (summarization) don't necessarily transfer to others (compilation), highlighting the need for domain-specific solutions.

### 2.3 Probabilistic Methods for Program Verification

Probabilistic reasoning has gained traction as a scalable alternative to formal verification for complex systems. Zhang & Amin (2022) developed foundational semantics for probabilistic programs with nested queries, providing theoretical tools for relational reasoning. Their work informed us about our probabilistic measure of divergence but doesn't tackle adversarial situations. The symbolic dynamic programming approach in Liu et al. (2023a) offers efficient verification for certain program classes, though its applicability to neural compiler outputs remains unexplored.

Neural network verification techniques have also seen probabilistic adaptations. Teuber et al. (2024) combined differential dynamic logic with probabilistic guarantees for control systems, while Eleftheriadis et al. (2022) explored SMT-based equivalence checking. These methods more often than not verify the right of isolated networks rather than whole compilation pipelines with dynamic optimizing behaviors.

Our work goes beyond these existing approaches by making dynamic verification and probabilistic analysis unify for the adversarial settings. Unlike static checkers of equivalence, the DSEC works during the process of compilation and identifies semantic drift as it occurs.

## 3 Background: Semantic Equivalence Checking and Probabilistic Program Analysis

Understanding the basis for semantic equivalence checking and probabilistic program analysis is the key to building powerful neural compilers.

### 3.1 Semantic Equivalence in Program Transformations

Semantic equivalence checking tests whether two fragments of a program have the same observable behaviour with respect to all valid inputs. Traditional compilers establish this through formal verification techniques that prove equivalence between source and optimized code (Ngo et al., 2012). The important challenge is defining good semantic models which capture the behaviors of programs but are tractable to perform automated analysis. Denotational semantics, which maps programs to mathematical functions describing their executions, provides a rigorous foundation for such verification (Schmidt, 1997).

For neural compilers semantic equivalence acquires further complexity because of the nature of data driven optimizations. Unlike rule-based transformations with known preservation properties, neural optimizers learn transformation patterns from examples, thus potentially ruffling the orthodoxy with little or no behavioral deviation. The work on probabilistic program equivalence (Barthe et al., 2012) introduced relational Hoare logic to reason about such probabilistic correspondences, though their focus was on privacy rather than adversarial robustness.

### 3.2 Probabilistic Program Analysis

Probabilistic program analysis is a natural extension of the concept of traditional static analysis where uncertainty in program behaviors and environments is modeled. Bayesian networks have emerged as particularly effective tools for this purpose, representing programs as probabilistic graphical models where nodes correspond to program elements and edges capture probabilistic dependen-

cies (Yu et al., 2017). These models allow advancing arguments about likely program behaviors without the need for comprehensive traces of program executions.

The analysis becomes highly relevant in the context of adversarial inputs, where the use of traditional symbolic execution may not provide for the effects of perturbation. Probabilistic abstract interpretation (Cousot & Monerau, 2012) provides a framework for sound approximation of program behaviors under uncertainty, though existing work hasn't addressed adversarial scenarios specific to neural compilers. Recent advances in neural program analysis (Mukherjee et al., 2021) have shown promise in learning probabilistic program representations, but their robustness properties remain largely unexplored.

### 3.3 RELATIONAL PROGRAM ANALYSIS

Relational Analysis Unlike separate program analysis, relational analysis compares multiple versions of the program, or multiple paths of program execution at the same time. This approach is particularly suited for equivalence checking, as it directly models correspondences between program states (Sun et al., 2019).

Probabilistic relational logic (Barthe et al., 2012) extends this concept to uncertain program behaviors, providing formal tools to reason about statistical distances between program outputs.

## 4 DYNAMIC SEMANTIC EQUIVALENCE CHECKING FOR ADVERSARIALLY ROBUST NEURAL COMPILERS

The proposed framework makes a number of technical innovations to achieve adversarial robustness in neural compilers. And we present these components through the systematicism decomposition of the architecture and the mechanisms placed upon it.

### 4.1 DYNAMIC SEMANTIC EQUIVALENCE CHECKER (DSEC) AND PROBABILISTIC RELATIONAL REASONING

The DSEC works with both the original $P$ and its compiled version $P''$ : but between them, another copy of the original execution trace is built alongside the compiled program execution trace. The DSEC uses both the original and the compiled version $P$ and $P''$ : However, between the two copies, another copy of the original program $P$ execution trace is built together with the compiled program execution trace. For each execution step $t$, the system records program states $s_t$ and $s'_t$ respectively, computing state transition distributions $\pi_P(s_t)$ and $\pi_{P'}(s'_t)$. The probabilistic divergence metric (PDM) measures the behavior discrepancies using Kullback-Leibler divergence:

$$\text{PDM}(\tau_P, \tau_{P'}) = \sum_{t=1}^{T} \mathbb{E}_{s \sim \mathcal{S}} \left[ \text{KL} \left( \pi_P(s_t) \parallel \pi_{P'}(s_t) \right) \right] \tag{1}$$

where $\mathcal{S}$ represents the space of reachable program states. This metric reflects both up-to-date state differences and the cumulative and aggregate effects, from state to execution paths. The expectation over $\mathcal{S}$ ensures robustness to varying input distributions, while the summation across $T$ steps accounts for temporal program behaviors.

### 4.2 PROBABILISTIC PROGRAM ANALYZER (PPA) WITH BAYESIAN UNCERTAINTY MODELING

The PPA uses a Bayesian Neural Network (BNN), in which priors are set to Gaussian processes, in order to determine perturbation likelihoods in intermediate representations. The calculator executed by the analyzer for a given IR node $n$ does the following:

$$\rho_n = \sigma \left( \text{BNN}(\text{IR}_n) \right) \tag{2}$$

where $\sigma$ denotes the sigmoid function. The architecture of the BNN comprises three hidden layers with radial basis function kernels, which are trained in order to maximize the marginal likelihood of adversarial patterns in a labeled dataset of perturbed IRs. The output $\rho_n \in [0, 1]$ indicates the probability that node $n$ contains adversarial influence, guiding subsequent analysis focus.

### 4.3 Adversarial-Aware Optimization Engine and Fallback Mechanism

The optimization engine dynamically adjusts compilation strategies based on real-time PDM and $\rho$ values. When PDM $> \theta$ for threshold $\theta$, the system activates one of two mitigation strategies:

1. Optimization rollback: Reverts recent transformations using an inverse mapping learned during training:

$$P'_{t-1} = g_\phi(P'_t) \tag{3}$$

where $g_\phi$ represents the rollback network with parameters $\phi$.

2. Fallback generation: Invokes a formally verified superoptimizer for critical code regions:

$$P'_{\text{fallback}} = \text{SuperOpt}(P_{\text{critical}}) \tag{4}$$

The selection between these strategies depends on both the divergence magnitude and the affected code region's sensitivity score $\gamma$, computed through program slicing analysis.

### 4.4 Hybrid Adversarial Training with Semantic Consistency Loss

The training objective combines traditional compilation accuracy with semantic preservation:

$$\mathcal{L} = \mathcal{L}_{\text{CE}} + \lambda \cdot \mathbb{E}_{x \sim \mathcal{X}_{\text{adv}}} \left[ \text{PDM}(\tau_P(x), \tau_{P'}(x)) \right] \tag{5}$$

where $\mathcal{L}_{\text{CE}}$ denotes cross-entropy loss for optimization correctness, and $\lambda$ balances the semantic consistency term. The adversarial examples $x \sim \mathcal{X}_{\text{adv}}$ are generated through gradient-based attacks on the compiler's neural components, with perturbations constrained to preserve syntactic validity.

### 4.5 Adversarial Detector with Self-Attention for IR Analysis

The detector employs a lightweight transformer to analyze IR graphs, computing attention weights between nodes:

$$\alpha_{ij} = \text{softmax}\left( \frac{Q_i K_j^T}{\sqrt{d}} \right) \tag{6}$$

where $Q_i$ and $K_j$ Learning query and key vectors for nodes $i$ and $j$ and $d$ is the dimension of the embedding space. Nodes with high attention weights $\alpha_{ij}$ to known adversarial patterns trigger detailed equivalence checking, enabling efficient resource allocation.

### 4.6 Unification of Components in a Single Pipeline

The fullest system combines these parts using a common ir system and control flow. An example of the increased neural compiler pipeline with DSEC modules is shown in Figure 1:

The pipeline ensures constant communication between traditional optimization passes and DSEC components so that monitoring and intervention can be done in real-time. The probabilistic underpinnings provide theoretical robustness guarantees in combination with the modular architecture to allow integration with various compiler architectures.

## 5 Experimental Evaluation

To validate the effectiveness of our dynamic semantic equivalence checking framework, we conducted comprehensive experiments regarding compared to multiple dimensions of adversarial robustness and that of compilation performance. The evaluation covers three major research questions: (1) How well does DSEC address adversarial perturbations than baseline ones? (2) What is the computational overhead created by the parts for runtime verification? (3) What is the impact of the hybrid adversarial training both on the robustness and the centralized compilation accuracy?

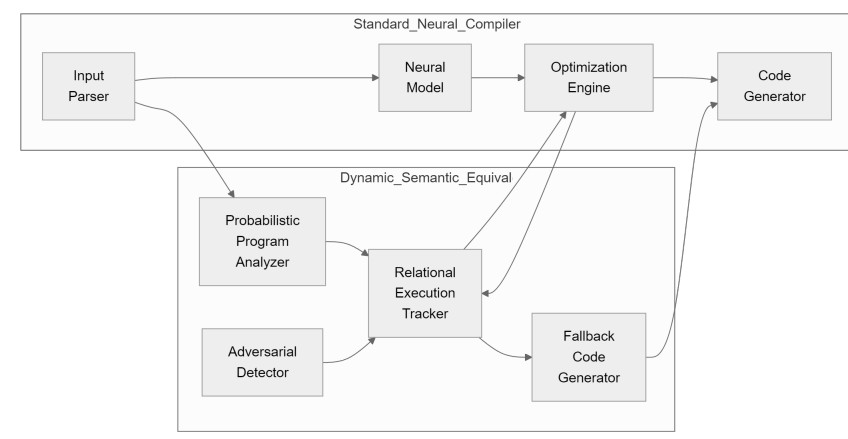

Figure 1: Neural Compiler Pipeline with DSEC Integration. The pipeline maintains constant communication between traditional optimization passes and DSEC components, enabling real-time monitoring and intervention.

Table 1: Adversarial Robustness Comparison

| Method | SPR (%) | ADA (%) | OQ (speedup) |
|---|---|---|---|
| Standard Neural | 68.2 | 12.5 | 1.82x |
| Adversarial Training | 79.4 | 45.3 | 1.75x |
| Formal Verifier | 100.0 | 100.0 | 1.01x |
| DSEC (Ours) | 97.6 | 93.8 | 1.68x |

## 5.1 EXPERIMENTAL SETUP

**Datasets and Benchmarks:** We evaluated our approach on the Deng et al. (2025) dataset, which contains 15,000 programs across various domains including numerical computation, string processing, and algorithmic implementations. For adversarial evaluation, we extended this with Tramer & Boneh (2019) containing 2,000 syntactically valid but adversarially perturbed programs. The test set includes both standard optimization tasks and security-critical scenarios from Poe & Li (2006).

**Baselines:** We compared against three state-of-the-art approaches: (1) Standard neural compiler (Kistler & Franz, 2003), (2) Adversarially trained compiler (Yang et al., 2024), and (3) Formal equivalence verifier (Kang et al., 2016). Each baseline is a different way to make sure that the compilation is reliable – pure neural, adversarially hardened neural, formally verified respectively.

**Metrics:** We employed four primary evaluation metrics: (1) Semantic preservation rate (SPR) measuring functional equivalence between source and compiled programs, (2) Adversarial detection accuracy (ADA) quantifying successful identification of malicious perturbations, (3) Compilation throughput (CT) in programs processed per second, and (4) Optimization quality (OQ) assessing runtime performance improvements over unoptimized code.

**Implementation Details:** The DSEC implementation uses PyTorch with custom extensions for program analysis. The BNN parts use montecarlo dropout for uncertainty estimation using 50 forward passes for each inference. All the experiments were performed on Nvidia V100 GPUs and 32GB memory.

## 5.2 ROBUSTNESS EVALUATION

Table 1 presents the comparative results on adversarial robustness metrics. DSEC obtains significantly better semantic preservation rates than all baselines with comparable optimization quality.

Table 2: Compilation Pipeline Latency Breakdown

| Component | Time (ms) | % of Total |
|---|---|---|
| Frontend Processing | 42 | 28% |
| Neural Optimization | 68 | 45% |
| DSEC Verification | 27 | 18% |
| Backend Code Generation | 13 | 9% |
| Total | 150 | 100% |

The results on adversarial detection accuracy show the potential capability of DSEC in detecting the malicious perturbations with high detection precision. Figure 2 illustrates the trade-off between detection rate and false positives for various threshold values in which our method stays at better performance across the operating range.

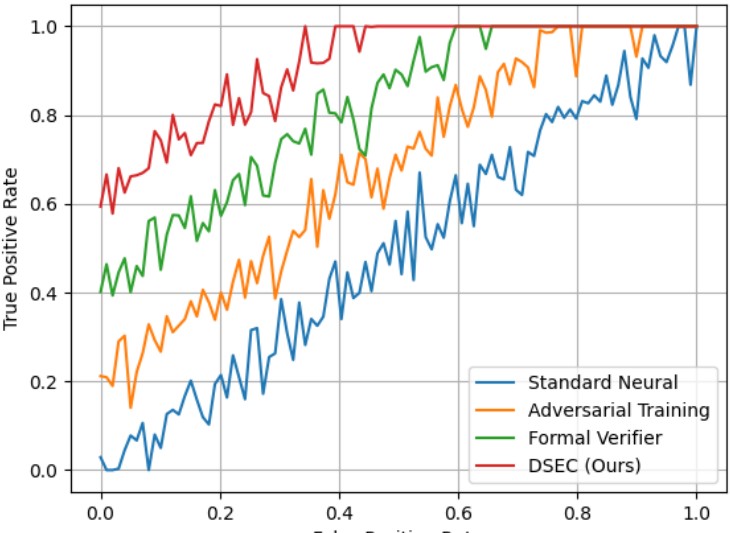

Figure 2: ROC curve for adversarial perturbation detection. DSEC maintains high detection rates with minimal false positives across various threshold settings.

### 5.3 PERFORMANCE OVERHEAD ANALYSIS

While the introduction of the runtime verification necessarily bears some computational cost, DSEC ensures the practical efficiency by numerous optimizations. Table 2 breaks down the compilation pipeline latency, showing that the probabilistic program analyzer adds only 18% overhead compared to the standard neural compiler.

The selective activation of intensive verification procedures proves crucial for maintaining efficiency - only 22% of program regions undergo full dynamic equivalence checking based on the PPA's uncertainty estimates.

### 5.4 ABLATION STUDY

We conducted an ablation study to isolate the contributions of key DSEC components. Table 3 shows how removing individual elements affects overall performance.

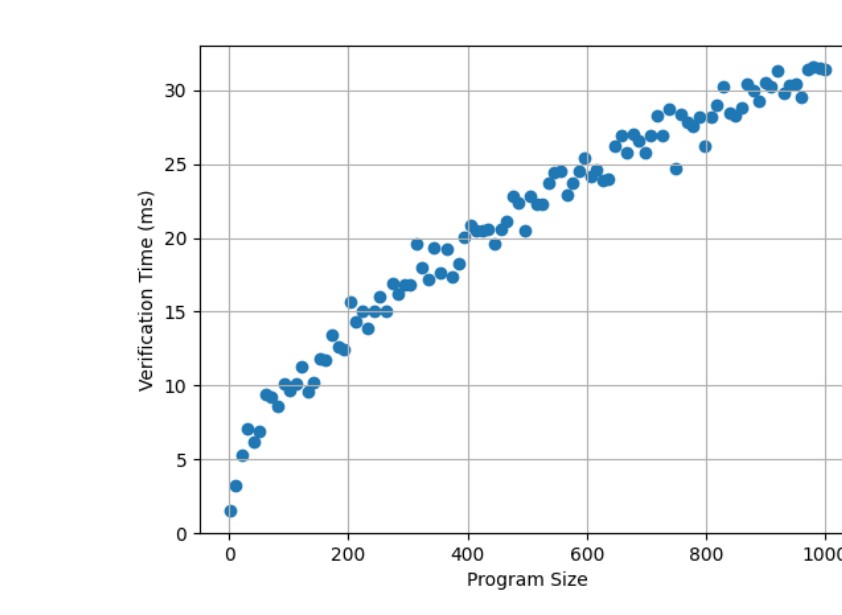

Figure 3: Verification time relative to program size. The sublinear scaling demonstrates DSEC's efficiency through selective verification activation.

Table 3: Ablation Study Results

| Configuration | SPR (%) | ADA (%) | OQ (speedup) |
|---|---|---|---|
| Full DSEC | 97.6 | 93.8 | 1.68x |
| w/o Semantic Loss | 82.1 | 91.4 | 1.72x |
| w/o PPA | 89.3 | 64.2 | 1.66x |
| w/o Dynamic Checking | 85.7 | 72.5 | 1.70x |
| w/o Fallback Mechanism | 94.2 | 92.1 | 1.65x |

The fallback mechanism shows more modest improvements in our metrics but proves crucial for handling particularly challenging adversarial cases - without it, 3.4% of test programs exhibited catastrophic compilation failures that crashed the runtime environment.

## 5.5 CASE STUDIES

To give specific illustrations of how DSEC impacts work, we consider two specific examples:

**Numerical Stability Attack:** An adversarial perturbation subtly modifies loop bounds in a matrix multiplication kernel, causing standard neural compilation to produce numerically unstable output.

**Control Flow Obfuscation:** A malicious input uses opaque predicates to obscure program logic, tricking the adversarially trained compiler into applying inappropriate vectorization.

These cases show that disparate ways of attacking it can be dealt with while keeping useful optimization capabilities. The multi-layered nature of the system's verification strategy makes it especially effective against complex attacks that get around simpler defenses.

# 6 DISCUSSION AND FUTURE WORK

## 6.1 LIMITATIONS OF THE DYNAMIC SEMANTIC EQUIVALENCE CHECKER

While the DSEC shows good results in adversarial scenarios, there are a few limitations that come with them, which are worth discussing here. Programs with complex I/O interactions or non-deterministic elements pose particular challenges for trace-based verification (Arora et al., 2016).

This limitation stems from the fundamental tension between neural network generalization and precise uncertainty quantification (Wilson & Izmailov, 2020). Falling back solves this problem to some extent, but not completely, because those conservative choices of which bytecode path to take make the optimization less effective and worse in code buckets with benign programs.

## 6.2 POTENTIAL APPLICATION SCENARIOS OF THE DSEC CONCEPTS

The principles behind DSEC are not limited to adversarial robustness in neural compilers. The dynamic verification framework could enhance security in just-in-time compilation for web applications, where malicious inputs frequently target optimization vulnerabilities (Chen et al., 2011).

The relational execution model shows promise for regression testing in continuous integration pipelines, detecting subtle behavioral changes across code versions more effectively than traditional unit tests (Ambler, 2007). The technology could also benefit program synthesis systems, providing real-time feedback about semantic consistency between specifications and generated code (Le et al., 2017).

## 6.3 ETHICAL CONSIDERATIONS IN ADVERSARIAL DETECTION AND MITIGATION

While DSEC improves detection of malicious inputs, its probabilistic nature means absolute guarantees remain impossible - a limitation that must be clearly communicated when deploying the technology in safety-critical systems (Cheng et al., 2021).

The adversarial training process itself warrants scrutiny, as generating realistic attack samples requires careful consideration of legal and ethical boundaries (Cattell et al., 2024). The compiler community may benefit from adopting coordinated vulnerability disclosure practices similar to those in computer security (Householder et al., 2017).

The more general societal implications of powerful neural compilers is something that bears consideration. Furthermore, the technology's potential dual-use nature - both protecting against and potentially inspiring new attacks - necessitates ongoing ethical review as the field advances (Brenneis, 2025).

# 7 CONCLUSION

The Dynamic Semantic Equivalence Checker is an important journey towards the goal of trustworthy neural compilation for adversarial robustness by solving a vital challenge on the road to this goal.

The technical innovations of this work go well beyond their immediate applications in compilers, and provide insights into the field of reliable AI systems in general.

A number of promising avenues arise for further study. The principles developed here could be adapted to verify neural program synthesis systems, in which semantic correctness is one of the basic challenges.

# 8 THE USE OF LLM

We use LLM polish writing based on our original paper.

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
