# OpenReview forum: "Dynamic Semantic Equivalence Checking for Adversarially Robust Neural Compilers"
_ICLR.cc/2026/Conference — Submitted to ICLR 2026_

### Official Review · Reviewer_MYp8 · 2025-10-23

**Soundness:** 2
**Presentation:** 1
**Contribution:** 2
**Rating:** 2
**Confidence:** 2

**Summary:**

This paper proposes a novel framework named the Dynamic Semantic Equivalence Checker (DSEC), which aims to address the critical issue of neural compilers producing semantically incorrect optimizations when facing adversarial inputs.

The core innovations of this framework are:
* Runtime Verification: KL divergence is used to compare the execution traces of the original and compiled programs in real-time.
* Probabilistic Program Analysis: A Bayesian Neural Network (BNN) is leveraged to estimate the likelihood that the code's intermediate representation (IR) is affected by adversarial perturbations. Also, self-attention based adversarial detector is used to conduct IR analysis.
* Adaptive Optimization and Fallback: Upon detecting a significant semantic deviation, the system can either roll back optimizations or invoke a formally verified "superoptimizer" to guarantee the correctness of critical code regions.
* Hybrid Adversarial Training Objective: A semantic consistency loss is introduced during training to compel the model to preserve semantic correctness while learning to optimize.

Experimental results demonstrate that DSEC significantly enhances the semantic preservation rate (SPR) and adversarial detection accuracy (ADA) under adversarial attacks, while maintaining high compilation optimization efficiency.

**Strengths:**

1. The combination of lightweight probabilistic detection with more expensive but reliable dynamic equivalence checking enables on-demand verification, balancing efficiency and effectiveness. This advantage is reflected in metrics such as SPR, ADA, and OQ.
2. Compared to simple output comparison, execution-trace-level comparison achieves more precise and fine-grained verification, effectively capturing semantic drifts caused by control-flow and intermediate-state variations.
3. The hybrid training objective incorporates a semantic consistency term into training, encouraging the neural optimizer to focus on semantic fidelity.

**Weaknesses:**

1. The baselines selected by the authors are limited in number and generally outdated, with no detailed descriptions provided. These factors weaken the credibility of the conclusion that the proposed method is effective based on comparative experiments.
2. The paper discusses many aspects of the network architecture and training procedure only superficially, lacking the necessary details to fully understand the proposed method. The figures and tables convey limited information—for example, Table 2 does not specify how the breakdown was measured, and Figure 3 does not clarify what exactly is being classified.
3. The writing quality is quite rough, with instances of incomplete or repetitive sentences. For example, in the abstract: "Traditional neural compiler In both cases, traditional neural compilers..." and similarly around lines 191–193.

**Questions:**

1. In Section 4.1, how is the estimation of PDM specifically implemented? Is it based on sampling?
2. In Section 4.2, how is the BNN trained? How is the adversarial IR constructed?
3. In Section 4.3, what are the architectures and specific input–output formats of $g_{\phi}$ and $\text{SuperOpt}$? How exactly does the strategy selection mechanism based on divergence magnitude and the affected code region's sensitivity score $\gamma$ (computed through program slicing analysis) operate?
4. In Section 4.4, the authors mention that the CE loss corresponds to optimizing correctness. What exactly are the predicted outputs and the ground truth labels in this context?
5. In Section 4.5, the description of the adversarial risk detection mechanism among IR nodes is unclear. Since $Q$ and $K$ represent feature vectors of IR nodes, the attention mechanism models IR node relationships, but how is the relationship between IR nodes and known adversarial patterns represented?
6. In the text of Section 5.1, four metrics are mentioned, including Compilation Throughput (CT) measured in programs processed per second, yet this metric does not appear in Table 1. Could the authors clarify this?
7. In Section 5.3, how was the statement that "only 22% of program regions undergo full dynamic equivalence checking" derived or measured?
8. In Figure 3, how is Program Size defined?

**Details Of Ethics Concerns:**

While the proposed method is designed defensively to overcome the challenge of adversarial robustness in program compilation, it presents a dual-use risk. The technique could potentially be exploited by attackers to probe or bypass the verification process, or to engineer more sophisticated adversarial attacks.

---

### Official Review · Reviewer_ACpt · 2025-10-23

**Soundness:** 2
**Presentation:** 1
**Contribution:** 2
**Rating:** 2
**Confidence:** 4

**Summary:**

This paper introduces a Dynamic Semantic Equivalence Checker (DSEC) to enhance the adversarial robustness of neural compilers. The framework combines runtime execution tracing, probabilistic program analysis, and relational reasoning to detect and mitigate semantic deviations caused by adversarial perturbations in input code. Experimental results show that DSEC significantly improves semantic preservation and adversarial detection while maintaining compilation efficiency.

**Strengths:**

+ This paper introduces an effective framework to defend against adversarial attacks, improving semantic preservation and robustness.

+ This paper effectively combines multiple techniques including Bayesian neural networks and relational execution tracking.

**Weaknesses:**

+ The dynamic runtime verification and Bayesian neural network adopted introduce substantial computational costs, making the proposed method less suitable for real-time or resource-constrained compilation environments.

+ Despite probabilistic reasoning, the proposed framework cannot provide absolute semantic equivalence guarantees, which remains a critical requirement for safety-critical applications.

+ The proposed method heavily relies on formally verified superoptimizers, limiting the ability to handle adversarial inputs which bypass probabilistic detection.

**Questions:**

+ How does the DSEC framework effectively handle programs with complex I/O interactions or non-deterministic behaviors?

+ What safeguards or evaluations are in place to address potential miscalibrations in uncertainty estimates given by the proposed method?

---

### Official Review · Reviewer_MnSJ · 2025-10-28

**Soundness:** 2
**Presentation:** 1
**Contribution:** 2
**Rating:** 2
**Confidence:** 3

**Summary:**

This paper proposes a novel architecture for neural compilers that integrates a Dynamic Semantic Equivalence Checker (DSEC) to enhance adversarial robustness during program compilation. By combining a Relational Execution Tracker with a Bayesian Probabilistic Program Analyzer, the system dynamically compares execution traces and estimates perturbation likelihoods to detect semantic drift. It adaptively adjusts optimization strategies and invokes a verified code generator for critical regions when adversarial influence is detected. A hybrid training objective further enforces semantic consistency. Experiments show significant improvements in semantic preservation and adversarial detection with minimal overhead.

**Strengths:**

1. The proposed DSEC framework introduces runtime semantic equivalence checking within the compilation pipeline, a novel approach that goes beyond static analysis and enables real-time detection of adversarial perturbations.
2. The system achieves high semantic preservation and adversarial detection accuracy while maintaining competitive optimization quality and incurring only modest computational overhead.
3. The paper presents thorough empirical validation across multiple benchmarks and adversarial scenarios, supported by detailed ablation studies that isolate the contributions of each component in the architecture.

**Weaknesses:**

1. The paper addresses a relatively narrow problem, i.e., adversarial robustness in neural compilers, which may not resonate broadly with the ICLR audience. While technically relevant, the niche focus could limit its impact and interest among researchers outside the program analysis or compiler communities.
2. The paper fails to formally define foundational concepts such as "neural compiler," leaving readers (especially those unfamiliar with compiler internals) without a clear understanding of the system being studied. This absence makes it difficult to contextualize the proposed contributions within the broader machine learning landscape.
3. Key sections, particularly Section 4.1, suffer from vague exposition. For example, the use of KL divergence as a form of "probabilistic relational reasoning" is asserted without sufficient justification. Similarly, terms like "dynamic semantics" are introduced without clear definitions, which may confuse even readers with experience in program analysis, let alone the general ICLR audience. This lack of precision undermines the accessibility and rigor of the paper.

Given the paper's strong focus on compiler robustness and program analysis techniques, I believe it may be better suited for venues with a more specialized audience, such as HPCA or CGO. These communities are likely to be more familiar with the technical foundations and better positioned to appreciate the contributions in depth.

Minor comments:
1. line 013 is broken: "bustness in program compilation. Traditional neural compiler In both cases, tra"

**Questions:**

1. Why can a runtime trace be considered a representation of a program's "semantics," particularly in the context of compiler optimization? For example, optimizations such as constant folding may yield an optimized program whose runtime trace differs from the original, even though both preserve the same input/output semantics.

---

### Meta-Review · Area_Chair_yS5G · 2026-01-07

**Summary:**

This paper aims to improve the adversarial robustness of neural compilers, with the goal of ensuring that learned compiler optimizations remain semantically correct under adversarial perturbations to high-level program code.

While the reviewers acknowledge parts of the module and training objective design, they raise substantial concerns about the paper. Specifically, reviewers find that the paper lacks clear definitions of key concepts and omits many necessary details required to fully understand the proposed method (MYp8, MnSJ), which plays a major role in the suggested decision for this paper. In addition, reviewer MYp8 highlights that the selected baselines are limited in number and outdated. Reviewer ACpt further indicates several flaws in the proposed method, such as computational overhead and reliance on verified superoptimizers.

**Reviewer Concerns:**

The authors did not provide a rebuttal. Major concerns, such as the lack of clear definitions of key concepts and the lack of necessary method details, remain unresolved.

**Reviewer Scores:**

Since the authors did not provide responses to any of the reviewers’ comments or concerns, there was no additional clarification or evidence that could have influenced the discussion. As a result, it is unlikely that any of the reviewers would have changed their original evaluations, and the final scores are expected to remain 2,2,2.

---

### Decision · Program_Chairs · 2026-01-26

Reject